# Patients’ Anticipation for the Pharmacies of Rural Communities: A Qualitative Study from Pakistan

**DOI:** 10.3390/ijerph16010143

**Published:** 2019-01-07

**Authors:** Muhammad Majid Aziz, Minghuan Jiang, Imran Masood, Jie Chang, Shan Zhu, Muhammad Ali Raza, Wenjing Ji, Caijun Yang, Yu Fang

**Affiliations:** 1Department of Pharmacy Administration and Clinical Pharmacy, School of Pharmacy, Xi’an Jiaotong University, Xi’an 710061, China; pharmajid82@yahoo.com (M.M.A.); tianji712@126.com (M.J.); jiechang@xjtu.edu.cn (J.C.); zhu1992220@163.com (S.Z.); yfyx_8312@163.com (W.J.); yangcj@mail.xjtu.edu.cn (C.Y.); 2The Center for Drug Safety and Policy Research, Xi’an Jiaotong University, Xi’an 710061, China; 3Global Health Institute, Xi’an Jiaotong University, Xi’an 710061, China; 4Shaanxi Center for Health Reform and Development Research, Xi’an 710061, China; 5Department of Pharmacy, Faculty of Pharmacy and Alternative Medicine, Railway Road Campus, Islamia University, Bahawalpur 63100, Pakistan; drimranmasood@gmail.com; 6Faculty of Pharmacy, Bahauddin Zakariya University, Multan 66000, Pakistan; alirazaabbasi82414@gmail.com

**Keywords:** qualitative study, patients’ anticipation, community pharmacies, rural population, Pakistan

## Abstract

*Background*: Community pharmacies are an integrated part of healthcare systems worldwide. In low and middle income countries like Pakistan, the paradigm of pharmacy practice is shifting from dispensing medicines to clinical activities. There are disparities in these practices according to location. Pharmacies in urban localities are better than those in rural areas. This qualitative study was conducted to explore patients’ expectations and current practices in rural pharmacies. *Methods*: A cohort of adult pharmacy visitors (aged > 18 years) that reside in rural community was selected. Consenting participants were recruited by purposive sampling technique until thematic saturation level was achieved. A total of 34 patients were interviewed. Face-to-face interviews were conducted using a semi structured interview guide. All the data were transcribed and used to originate the themes. *Results*: On analysis, a total of 20 themes were obtained. Sixteen themes pronounced the current provided services. Four themes provided some suggestions for the development of better pharmacies. On call services to provide medicines, limited free extended pharmacy services, interest in patients’ wellbeing, appropriate referral, vaccination, free medical camp, medical services at home, first aid, and counseling were appreciated by patients. Patients stated that medicines are inappropriately stored in unhygienic conditions, prices of medicines are comparatively high, and medicines are substandard. Unavailability of medicines, inept dispensing, limited staffing with poor knowledge, limited working hours, and quackery promotion are challenges in rural pharmacy practice. Patients say that non marginal pricing, informative services, new legislation, and proper vigilance by officials can improve the pharmacy services in rural communities. *Conclusions*: Patients alleged that rural pharmacies perform deprived practices. To improve service, new legislation and the proper implementation of existing law is needed.

## 1. Introduction

Community pharmacies are an integrated part of the healthcare system globally [1]. Community pharmacies play a very important role in selling medication and the provision of patient care services [2]. During the last three decades, philosophies of pharmacy practice have changed globally from apothecary to effective therapy, with ensured patient safety and a reduction in treatment cost, supported by liaison with other healthcare providers [3]. In previous decades, various novel community pharmacy services have been successfully established to provide optimal health outcomes for many clinical areas [4]. Contemporary community pharmacy practices exhibit a paradigm shift from dispensing medicines to clinical activities [5]. 

Patient views are valuable predictors for evaluating health service management and healthcare delivery systems [6]. Patient perception is also an important indicator by which to judge community pharmacy services [4,6]. This feedback from patients helps to improve pharmacy services [6,7]. Community pharmacies can develop strategies to improve their services on the basis of patients’ preferences [8,9].

Similar to many countries, community pharmacies are the most accessible part of the medical service in Pakistan [10,11]. In Pakistan, the private sector makes up the predominant proportion of health care system. Most pharmacies are also in the private sector [10,12,13,14,15,16]. There are more than 80,000 community pharmacies in Pakistan [11]. These pharmacies have geographical diversity in their distribution, and disparities in their performance with regard to location [12,13]. The absence of pharmaceutical care through pharmacies has been reported. The legal licensing requirements are not fulfilled. The overall practices are unsatisfactory, due to unqualified supervision [12,13,14,15,16]. The practices and services of urban pharmacies are considered better than those of pharmacies in rural areas [10]. No study has yet represented the performance of pharmacies rural community. No comparative study has demonstrated the working of pharmacies in rural and urban areas. 

This study aimed to explore the needs of rural patients and the current services of rural pharmacies as compared to urban pharmacies. To best of our knowledge, this is first study to reflect patients’ views about community pharmacies in rural areas of Pakistan.

## 2. Methods

### 2.1. Study Design

To determine the exact needs of patients, a qualitative method was applied. This design was self-sufficient and flexible, for deep exploration of patients’ needs and expectations. This method also originated a variety of ideas and viewpoints to found unexposed areas [17,18,19]. This method sanctioned detailed investigations of participants. For this purpose, the interviews of patients were conducted between August and September 2016.

### 2.2. Study Instrument

A semi-structured interview guide (Appendix A) was developed from previous studies [4,7,8,20,21,22,23,24,25,26,27,28,29,30,31]. The interview guide had two portions: one focused on current service, and the second on suggestions for service improvement. This interview guide focused on pharmacy organization and operations, standard of provided medication, staffing, dispensing service, counseling service, and suggestions for improvement. Experts assessed the validity and reliability of the guide according to the nature of the focused community. Before data collection, the guide was further assessed by a pilot study. For this purpose, face-to-face interviews of 2 study participants were conducted, and the guide was modified partially according to obtained feedback. The interview guide was composed of 13 open ended questions.

### 2.3. Study Setting

Study sites were community pharmacies in three divisions—Bahawalpur, Dera Ghazi Khan, and Multan—known as south Punjab, Pakistan [32]. The total population of this area is 34,747,064 individuals [33]. According to the World Bank, about 64% of the total population of Pakistan is rural [34]. The ratio of rural population in Punjab is comparatively high, estimated at more than 68.70% of total population [35].

### 2.4. Selection Criteria of Participants

Native adult (aged > 18 years) pharmacy visitors of rural community were selected. The rural population was defined according to the criteria of the primary & secondary healthcare departments of Punjab [36]. To ensure a good level of participant awareness about different pharmacy settings, an inclusion criterion was defined.

### 2.5. Inclusion and Exclusion Criteria

#### 2.5.1. Inclusion Criteria

Only patients which met the following criteria were included:Patients have visited more than 10 different urban pharmacies.Patients have a valid prescription from a registered medical practitioner.Patients declare no competing interest to pharmacy owners/staff.

#### 2.5.2. Exclusion Criteria

Patients that met the following criteria were excluded:Patients with any conflict of interest or any relation to owner of pharmacy.Patients visiting pharmacy for over-the-counter (OTC) medicine.Patient under the age of 18 years.

### 2.6. Sampling and Data Collection

A purposive sampling technique was applied in the selection of pharmacies. To select patients from these pharmacies, we applied a simple random selection procedure. In qualitative research methods, a combination of sampling methods is considered more appropriate, purposeful, and ensures maximum variation in the sample [37]. Patients which met the selection criteria were requested to participate. The consenting participants were recruited until thematic saturation level was achieved. The interviews were conducted according to participants’ feasible time and location, with complete freedom of expression. All dialogues were audio-recorded and complete field notes were prepared. Probing questions were asked when needed. The duration of interviews was 30–45 min. Saturation level was achieved at the 30th interview. However, 4 more patients were interviewed to reconfirm saturation level.

### 2.7. Data Analysis

For complete understanding, interviews were performed in local languages (Urdu, Punjabi, and Saraiki). Before transcribing; two qualified and skilled researchers (I.M. & M.A.R.) carefully translated all interviews into English. No grammatical correction was done during the process of transcription, as Pakistani participants were not proficient in English. To determine inter-rater reliability, another researcher listened to the recording and reviewed the transcribed copies. The contents were reanalyzed to identify regularities and outlines within the data [38]. Grammatical errors in quotations were then corrected, and transcription errors were eliminated upon the discussion and mutual consent of the three researchers concerned (M.M.A., I.M. & M.A.R.).

The data was coded and categorized to develop themes. Each line of transcript was analyzed by the first author (M.M.A.) to identify the expected themes. To guaranteed precision, contents and themes were also verified twice by two co-authors (W.J. & M.J.). Microsoft Excel was used to summarize the demographic characteristics of the participants.

### 2.8. Ethics

The study design and protocol were approved by the research Centre for Drug Safety and Policy (CDSP-16-PHD1-P4), after the ethical permission of Xi’an Jiaotong University (Ref # MR102-15/Phar) and Pharmacy Research Ethics Committee (PREC) at the Islamia University Bahawalpur, Pakistan (Ref # 67-2015/PREC). In addition, written and verbal consent was obtained from participants (Appendix A) and pharmacy retailers (Appendix A). The identities of participants and pharmacies were anonymized. Identification numbers were used in data collection and monitoring. All participants were informed of the study purpose.

## 3. Results

A total of 34 patients from licensed pharmacies were interviewed. All respondents were male. Most of them (64.7%) were aged above 40 years. More than half (58.9%) had education below the bachelor level. About half (55.9%) participants had a monthly income of 15,001–30,000 PkR (approximately 122–243 USD). Many (26.5%) suffered from respiratory infections (Table 1).

### 3.1. Current Services

Data analysis determined the following eleven major themes:

#### 3.1.1. Inappropriate Unhygienic Storage

Patients claimed unhygienic and improper medicine storage. The infrastructure of the pharmacies did not ensure medication safety:
“The pharmacies shops are not more developed than a local grocery or departmental stores ...... the rakes are broken and uncovered ...... no glass cover of rakes ...... many medicine are kept in big open containers ....... Cattle farming are common here and their dung is spread around ....... How the medicine can pure, safe and efficacious?”*(P 4)*
“The humming flies welcome the patients ever ...... absence of front glass door ...... presence of dust over the packing and blisters of medicines is common” *(P 20)*
“The floor of pharmacies in cities are blazing, flat and decorated while in our locality dilapidated and un even floor can be seen”*(P 11)*
“Medicines are not stored in clean racks, the shops of pharmacy is also not clean well, staff don’t have clean dresses”*(P 33)*

Moreover, a participant exposed improper cold chain maintenance:
“My doctor told me insulin without maintenances of cold chain loss their efficacy ...... pharmacy have no refrigerator ...... 18 h electricity breakdown is common in villages ...... no alternative power supply ...... purchasing such medicines from these pharmacies is waste of money ...... I prefer to purchase the my own medicines and eye drops for my mother from city”*(P 7)*

#### 3.1.2. Having Interest in Patient’s Wellbeing and Counseling

The owner/staff of pharmacy seem interested in the well being and counseling of patients:
“When patients are not much, provide enough time to discuss disease and therapies in a very polite way ...... in rush to deal other patients, ignore some time ...... in urban pharmacies ignorance is often ...... on request staff of urban pharmacies referred us again to concern doctors”*(P 6)*
“He is very friendly to me and almost to all patients ...... many time told me about the harm of smoking …... spread of diseases like TB ...... eating habits and importance of regular exercise to control blood pressure and diabetes”*(P 1)*
“Provides the significant counseling and deals the health queries in a good way”*(P 26)*
“Medication counseling and instructions regarding wellbeing are significantly provided specially in chronic diseases”*(P 31)*

#### 3.1.3. Appropriate Referral

Patients were contented about the referral services by the staff of pharmacies:
“As most of the people in our village are not aware of specialized doctors and their addresses in city ...... finding a proper doctor for a specific disease and approaching to concern doctor is a difficult task for the attendants of patients specially in the case of emergency ....... We contact to the owner of pharmacy for this purpose ....... He manages every thing ...... provide us the exact information”*(P 22)*
“He referred me to a doctor for my cardiac problem ...... my wife to gynecologist and my son to pediatrician ...... we find these doctors very cooperative and competent ...... this services by him save our time and money”*(P 29)*


#### 3.1.4. Vaccination Point

Some pharmacies are vaccination points of rural communities:
“Government vaccinator come to pharmacy ...... people of village are informed prior about the vaccinator visit ...... children and women got free vaccinations according to schedule”*(P 26)*
“The owner of pharmacy is like a family member of each house ...... contact to head of each family ...... load speaker of mosque is used ...... government vaccinator stay a complete day at the pharmacy ...... people visit for vaccination according to their feasibility”*(P 31)*

#### 3.1.5 Charging High Prices for Medicines

The participants argued about the prices of medicines. They said prices are high compared to urban pharmacies, and no discount to patients is offered by rural pharmacies:
“Five to ten percent discount is common in urban pharmacies ....... up-to 15% discount can find there due to business competition ...... facilitate patients without request ... having a single pharmacy in villages ...... can’t get any discount even by request. Simply (a business) monopoly”*(P 5)*
“The prices of medicines are personal description here ...... huge difference of medicines prices in urban and rural pharmacies”*(P 12)*
“In some cases, the written prices of medicines are much higher than original selling prices. In city, these medicines are available at 1/5th prices of written price e.g., folic acid preparation. But pharmacies in this area charge it as written on pack”*(P 8)*
“Very high prices of medicines are charged here”*(P 34)*

#### 3.1.6. Providing Substandard Medicines

Very dangerous pharmacy practices were exposed by patients’ comments. Illegitimate and substandard medicines are being sold to human through these channels:
“Veterinary medicines are freely decorated in the pharmacies ...... injecting and selling veterinary medicines to human beings without discrimination”*(P 9)*
“Mostly substandard and counterfeit medicines are sold through rural pharmacies”*(P 32)*
“Whenever I purchase medicines for myself/family/relative/friend, must check its expiry dates ...... Usually expiry dates are just one to 2 month latter ...... medicines in these pharmacies are near expiry”*(P 13)*

#### 3.1.7. Unavailability of Medicines

Patients reported challenges regarding the availability of required medicines. Patients need to wait to find required medicines. Only few medicines can be obtained in a single visit:
“The medicines of common illness like fever, flue, cough and diarrhea can obtain in a single visit .............. the medicines for other diseases need multiple visits or a visit to city”*(P 14)*
“Required medicines or a suitable substitute are arranged ...... a prior intimation, ensures the availability of formula milk for babies”*(P 18)*
“All medicines are not available every time .......... 1–3 days prior request is needed to get exact dose and dosage of cardiac and diabetic medicines .......... alternative (herbal and homeopathic) medicines are abundant in pharmacy”*(P 21)*
“I need epilepsy medicine for my son ....... pharmacy is on the way to my work place ....... I inform the owner 2 days ago and he manages medicines at the time”*(P 3)*
“Only few medicines are available that are used in their quackery practices ....... most medicines of multinational companies are rare”*(P 32)*

In addition, a co-morbid patient suffering from cardiac and diabetic disorders said:
“Medicines of chronic diseases of my routine treatment can be get after 1 or 2 days—prior information to owner of pharmacy”*(P 10)*

#### 3.1.8. Inept Dispensing

The process of medicine dispensing was on a completely non-scientific basis. The wrong doses and dosages of medicines are being dispensed.
“In many cases the prescribed dose of drug is not available ...... high or low doses are dispensed .......... advice to break a single tablet or take 2 tablets”*(P 19)*
“Different dosage forms are dispensed rather than prescribed/required like omeprazole tablets instead of its sachet”*(P 16)*
“Usually biological/therapeutic equal medicines are dispensed to fulfill a prescription. Mostly these are replaced by some cheap medicines”*(P 1)*
“Only yellow colored dispensing envelope is used ...... medicines from a large container are poured and provided ...... patient is don’t know name of dispensed medicine” *(P 7)*
“Sometime, medicines are dispensed by underage school students”*(P 9)*

#### 3.1.9. Inadequate Staffing with Poor Knowledge

The number of staff members seems insignificant to accomplish the necessary jobs. They also have a low level of technical training and formal education.
“At every visit a new employ is seen ...... no concept of permanent staff ...... students of schools at evening come to learn this business …... these students cleans the store and sometime dispense the medicines”*(P 9)*
“Only owner is doing all duties ...... don’t have any other employ”*(P 15)*
“No staff member and owner of pharmacy have education more than 10 years of formal schooling”*(P 21)*
“The owners/staff of pharmacies provide misinformation about the use of medicines ...... many time doctor asked “Who told you this wrong information about disease/ medicine use” ...... have very poor scientific knowledge ...... don’t have any technical diploma”*(P 17)*

#### 3.1.10. On-call Services Provide Medicines

A good service to support the patients’ needs can be found in rural pharmacies. Patients consider this service a blessing.
“At night, villagers don’t have any other health facilities ...... mobile numbers of owner is given at board to contact in emergency ...... almost all the population of community avail this facility”*(P 12)*
“Its blessing, pharmacy remain open in public holidays or festivals …... if pharmacy is close, make a contact through mobile ....... owner never refuse our request of medicines”*(P 1)*

#### 3.1.11. Limited Free Extended Pharmacy Services

Some pharmacies are equipped with fundamental scientific devices to fulfill basic needs of patients. Limited free extended pharmacy services are provided to patients:
“Weighing machine, body thermometer, blood pressure equipments at pharmacy are totally free for all....... provide the facility of blood pressure measuring at pharmacy and home”*(P 6)*
“I can’t afford personal gluco-meter ...... this device is available at pharmacy ...... my blood glucose level is monitored easily ......... only the cost of strips is charged ...... helpful to manage my glucose”*(P 10)*

#### 3.1.12. Arrangement of Medical Camps

Medical camps are organized at rural pharmacies, which are very helpful for poor patients:
“Consultant physician come to pharmacy on weekly basis ....... patients who can’t afford travelling cost or can’t afford travelling due to poor health conditions, get health services at door steps”*(P 24)*
“Eye camps are usually organized by ophthalmologists or some individuals on charity basis at rural pharmacy ....... patients get the free of cost medicines and advices”*(P 27)*
“Free medical camps at pharmacies are occasionally organized ....... helpful for underprivileged population of rural areas”*(P 30)*

#### 3.1.13. At Home Medical Services

Pharmacies provide health services at home in villages:
“We don’t have any hospital here ........ no doctor is available ......... in any illness, we just call to pharmacy owner ...... he is available at cell every time ....... he visits for patient check up at home ...... advices the medicines up-to his competencies or guide for further steps for patient care ...... in our locality it is helpful for family health specially for women”*(P 23)*
“In many minor ailments like diarrhea, fever, cough and flue …... we contact him through cell .... tell him the patients conditions and he send us the medicine at home …... some time he come to check the patients at home”*(P 28)*

#### 3.1.14. First Aid Point

Pharmacies of rural communities are the best point of first aid, as mentioned by patients:
“In our daily life, during working in field …... for minor injury pharmacies provide the first aid and necessary medicine”*(P 25)*
“Provision of on time first aid by pharmacy, saves the many lives”*(P 26)*
“Satisfactory first aid facilities are provided by pharmacies …... suitable for the people of every field of life”*(P 31)*

#### 3.1.15. Limited Working Hours

Patients seem dissatisfied about very limited working hours of pharmacies.
“The working hours of pharmacy is ranged between sunrise and sunset regardless of summer and winter”*(P 2)*
“In unexpected weather condition like rain, medicines can’t be obtain ....... faraway urban pharmacy is last resort ...... owner itself belongs a faraway river side village”*(P 10).*

#### 3.1.16. Promoting Quackery 

Patients seem discontented by the behavior of pharmacy owners. Patients perceive that owners of pharmacies are promoting quackery in the villages:
“Owner behaves like a doctor ...... prefer patients must get his advice for medication, directly ...... sometime he advices patient for their medication change ...... also change the prescribed medicines with cheap substitutes ...... attitude is much better to patients seeking direct medication advice” *(P 5)*
“The owner of pharmacy claims “same medicines are advised by him and doctor”.......... pay much intention to the patient without prescription ......... found variation in the prices of same medicine according to prescription nature ... higher prices on another doctor prescription and lower on his direct advice”*(P 17)*

### 3.2. Suggestions

Data analysis determines the following four major themes:

#### 3.2.1. Scheduling Non-marginal Pricing

Accurate pricing can ensure there is no difference of prices between urban and rural pharmacies. Patients suggested scheduling a non-marginal pricing system.
“At the time of price decision, government authorities must consider the rural, underprivileged population of rural areas. The actual selling prices must printed on pack”*(P 12)*
“Most of rural population is not aware of this difference between written prices and actual retail prices ....... margin between these prices should minimize”*(P 8)*

#### 3.2.2. Developing Informative Services

Patients expected informative services. Personal engagement of the pharmacy owners and availability of scientific literature can promote health awareness:
“In rural areas the social media is not commonly used ....... lacks of public health awareness in communities .... accesses to health care is limited ......... pharmacy owner can effectively promote campaigns related to health promotion and to control the prevalence of infectious disease like currently Congo virus prevention campaign”*(P 18)*
“The available literature about medication use, life style modification for many chronic diseases like diabetes, hypertension is helpful ......... such a literature is rare in rural pharmacies ...... Similar to urban pharmacies, posters and brochure at sale counters will be helpful for the well being of rural population”*(P 10)*

#### 3.2.3. Urgent Need of New Legislation and Implementation

Patients perceived that new rules about the working and working hours of pharmacy could strengthen the pharmacy services. Good infrastructure should be required for the licensing of a pharmacy.
“The authorities should define the working time for pharmacies in rural areas and ensure it by enforcement”*(P 2)*
“The concern officers authorities didn’t approve pharmacies with improper infrastructure”*(P 4).*

#### 3.2.4. Proper Vigilance by Officials

Health officials of every rank can reinforce the law by proper visits. Patients recommended a uniform vigilance system:
“Only officials and government authorities can improve the pharmacy services in rural areas ...... faraway areas are neglected by them .......... pharmacies of countryside are rarely visited”*(P 19)*
“In the cities there is the chance of visits by high officials. Therefore, officers of low ranks are much attentive ......... pharmacies and health services of are rural areas unnoticed by staff and authorities ...... vigilance system in villages should strengthen”*(P 15)*
“Pharmacies in villages are death house ...... need more vigilance and strict control ... government and authorities must pay immediate intentions”*(P 32)*

## 4. Discussion

Evidence indicates that overall performance of pharmacies is poor. Major mechanisms of drug storage and dispensing are improper. Staff promote quackery and have limited scientific knowledge for therapeutic counseling. Working hours of pharmacies are limited. Medicine prices are high and quality is low. The reservations of patients as to the performance of pharmacies endorse the findings of previous studies [2,7,8,20]. Pharmacies in the rural settings of India and Pakistan have comparatively non-prolific services, due to a lack of market competition and arrhythmic distribution of pharmacies [11,16,39]. The comments of patients indicate that pharmacies also deviate from the drug laws of Pakistan [11,15].

Similar to previous findings from Cameroon, Karachi, and Rawalpindi, this study reflects inadequate drug storage facilities [15,40]. Proper drug storage facilities preserve the quality of medicines and reduce the overall cost of cure [15,16,41]. In 2003, the World Health Organization (WHO, Geneva, Switzerland) recommended guidelines for good storage practices [42]. This unlawful practice, coupled with improper shelves without glass covering, unhygienic conditions, and lack of cold chain has also been reported in some other studies of Pakistan [11,13,15]. As compared to the pharmacies of urban setting, the prices of medicines are high. The pharmacies did not provide any discounts to the patients. This practice is common in India and Pakistan [43,44]. 

In addition, availability and quality of medicines is fragmented. Similar to previous reports, this study revealed the use of veterinary medicines in human [45,46]. Expired medicine was a major concern of patients. The sale of expired medicine is also seen in Pakistan [46]. The unavailability of medicines rather than the treatment of fever, flu, cough, and diarrhea is a big problem in rural areas [13]. This study depicts monopolies of pharmacies in villages. Very low numbers of pharmacies in villages are a major reason for this. Usually pharmacies are clustered near government hospitals or located in markets. No legislation of Pakistan controls the unequal distribution of pharmacies [10]. Therefore, high prices of medicines are charged, and patients are compelled by the behavior of the retailer to prefer quackery. Some pharmacies in Pakistan are openly engaged in quackery [43,47]. In short, this study raises very serious allegations, like selling veterinary medicines to humans and promoting quackery. Keeping in mind the probability of such type of responses, or any other severity of opinion: the nature of society and the research method, only the patients that don’t have any competing interest to these pharmacies were selected. In rural areas of Pakistan, people have some business interests or mutual conflicts [48]. In addition, in Pakistani communities, especially in rural populations, social norms do not allow the participation of women in social, academic, or research activities [49].

The dispensing practices in rural pharmacies are very dissatisfactory. In general practice, 76% of Pakistani doctors dispense medicines of unknown compositions to patients [50]. Inappropriate drug dispensing causes the waste of health resources, morbidities, and mortalities [51]. Good dispensing practices ensure the delivery of right medicines to patients by accurate selection, appropriate packing, and proper labeling. This study reveals far worse dispensing practices than previous findings from urban areas of Pakistan [14,16]. As in a study from Mexico, the participants of this study reported the use of broken or crushed tablets. When the prescribed or required dose is unavailable, medicines, especially tablets, are crushed and cut in half to obtain the specific dose. The therapeutic window of enteric coated and sublingual medicines is altered, and the risk of carcinogenicity may increase by splitting tablets [52].

Limited free extended pharmacy services provided to patients is a real need of Pakistani rural communities. These services help to manage chronic disease [53]. Referral services and pharmacies serving as vaccination point were also explored in this study. This study also reflects the interest of pharmacy staff in the wellbeing of patients. They provide enough time to patients for the discussion of disease and therapy. However, the patients thought they had very tenuous scientific knowledge and very low formal education. The supervision of unqualified persons in the community pharmacy system of Pakistan is a major challenge [1,2,16,54].

Patients suggested non-marginal prices of medicines. The margin between written prices and actual selling prices only increases the profit of pharmacy owners. This difference makes medicines unaffordable for poor patients [43,44]. The participants recommended the role of pharmacies in the promotion of public health. In low and middle income countries, private pharmacies are the first channel of health care to population. Pharmacies in rural communities are ideally located to promote public health. They can successfully promote prevention campaigns of infectious diseases [9,15]. 

The participants of this study wished that there could be a minimum standard according to community requirement. The law of Pakistan still has no jurisdiction over the working hours of pharmacies [55]. The participants also suggested that proper vigilance of pharmacies was important. To improve the services of pharmacies, proper vigilance by drug inspectors is very necessary. Drug inspectors do not visit pharmacies according to regulations [11,12].

## 5. Conclusions

The overall performance of rural pharmacies was not rated highly by patients. The structure and services do not completely meet the basic needs of patients. The educational level of staff is also questionable. The practices are awfully illegitimate and need urgent correction. To improve service, new legislation regarding the distribution of pharmacies, and the proper implementation of existing law for good pharmacy practice is needed.

## 6. Limitations

Firstly, this report shows only the views of consenting patients from pharmacies of rural communities in the Multan district. The results cannot be generalized to the whole country. Secondly, this study did not provide information on the participants’ severity of disease. Thirdly, all participants were male, due to the socio-cultural norms of the study setting. Fourthly, only patients visiting pharmacies to fill a valid prescription from a registered medical practitioner were included in the study of their consciousness and priorities with regards to health and wellbeing. The views of patients visiting for OTC drugs may vary.

## Figures and Tables

**Table 1 ijerph-16-00143-t001:** Social demographics of study participants.

Characteristics	Category	Frequency	(%)
Age	18–30	4	11.8
	31–40	8	23.5
	41–50	13	38.2
	<50	9	26.5
Gender	Male	34	100
	Female	0	0
Education status	Below or matriculation	11	32.4
	Intermediate	9	26.5
	Bachelor	5	14.7
	Master	7	20.5
	Higher	2	5.9
Monthly income in Pakistani rupees (PkR)	15,001–30,000	19	55.9
	30,001–45,000	7	20.6
	45,001–60,000	8	23.5
Disease (for which participant visited pharmacy) *	Cardiac diseases	1	2.9
	Diabetes mellitus	5	14.7
	Gastrointestinal diseases	4	11.8
	Orthopaedic problems	8	23.5
	Respiratory infections	9	26.5
	Urinary tract infections	4	11.8
	Other	3	8.8

* 2 patients were co-morbid, only the disease for which they currently visited pharmacy was included.

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
