# Peer review of "Patients’ Anticipation for the Pharmacies of Rural Communities: A Qualitative Study from Pakistan"

_ijerph, 2019, doi:10.3390/ijerph16010143_

Round 1
Reviewer 1 Report
Thank you for giving me the opportunity to review the re-submitted version of this article. The quality of the manuscript was improved, but several minor concerns exist in this manuscript. Comments are listed below.
Methods:
4. L93: Are there no exclusion criteria for this study?
Previous Comment: According to the inclusion and exclusion criteria, how to select the individuals who participated in the study? Did the authors continuously recruit the participants?
Author’s response: Corrected, please see the line #112.
Additional Comment: I thought that the authors revised the line #109 (line #112 is about recording), but it was still unclear. How did the authors recruit the participants “randomly”? It should be very important point to consider the reliability of the study results.
Author Response
Dear reviewer,
We are very grateful to you for providing a highly professional review and very constructive comments. In this revision, we have tried to improve our manuscript according to your valuable comments and suggestions. According to journal's instruction, We have highlighted our insertion in manuscript by blue color and deletion by red color in "Revised Manuscript with Track Changes".
Thanks again for your valuable time. We believe that by your precious comments, now current version of manuscript is much improved. We hope that you will find it is worthy and acceptable for publication.
Point 1. Additional Comment: I thought that the authors revised the line #109 (line #112 is about recording), but it was still unclear. How did the authors recruit the participants “randomly”? It should be very important point to consider the reliability of the study results.
Response: Extremely very sorry for inconvenience in the line number. We think, this is the difference in the copy of track change and original one. For this mistake; we are extremely very sorry.
To clear the point of randomization, we sufficiently explain it. Please see the line # 109-113 in "Revised Manuscript with Track Changes". A reference from a peer review publication is also added to strengthen it . We hope, the point is sufficiently cleared with explanation and evidences.
Best regards
Yu Fang (Corresponding Author)

Reviewer 2 Report
Dear Authors,
Manuscript ID: ijerph-404143-v1 entitled “Patients’ Anticipation for the Pharmacies of Rural Communities: A Qualitative Study from Pakistan” by Muhammad Majid Aziz, Minghuan Jiang, Imran Masood, Jie Chang, Zhu Shan, Muhammad Ali Raza, Wenjing Ji, Caijun Yang and Yu Fang presents interviews with patients about community pharmacies in rural area in Pakistan.
The Authors extended the research and increased the number of patients included in this study (from 21 to 34 persons). The publication is more valuable. The manuscript is generally well written.
I have also some small suggestions in order to improve paper, which are the following:
L. 52: predictorto – predictor to
L. 55: strategiesfor theirbetter – strategies for their better
L. 56: ofmedical – of medical
L. 126: was – was (please standardize the font)
L. 138: % – % (please standardize the font)
References: from 5 ?????
L. 422: please keep the order of the literature (from 1).
Best regards,
Author Response
Dear Reviewer ,
We are very grateful to you for providing a highly professional review and very constructive comments. In this revision, we have tried to improve our manuscript according to your valuable comments and suggestions. According to journal's instruction, We have highlighted our insertion in manuscript by blue color and deletion by red color in "Revised Manuscript with Track Changes". In addition, formatting changes are also highlighted on the right side of manuscript in "Revised Manuscript with Track Changes" Thanks again for your valuable time. We believe that by your precious comments, now current version of manuscript is much improved. We hope that you will find it is worthy and acceptable for publication.
Point 1. L. 52: predictorto – predictor to
Response: Corrected as advised, Please see the line # 52 in track changed copy.
Point 2. L. 55: strategiesfor theirbetter – strategies for their better
Response: Corrected as advised, Please see the line # 55 in track changed copy.
Point 3. L. 56: ofmedical – of medical
Response: Corrected as advised, Please see the line # 56 in track changed copy.
Point 4. L. 126: was – was (please standardize the font)
Response: Corrected as advised, Please see the line # 130 in track changed copy.
Point 5. L. 138: % – % (please standardize the font)
Response: Corrected as advised, Please see the line # 142 in track changed copy.
Point 6. References: from 5 ?????
L. 422: please keep the order of the literature (from 1).
Response: Corrected as advised, Please see the line # 425-541 in track changed copy.
Best regards
Yu Fang (Corresponding Author)
